# Proteomics in Allopolyploid Crops: Stress Resilience, Challenges and Prospects

**DOI:** 10.3390/proteomes13040060

**Published:** 2025-11-11

**Authors:** Tanushree Halder, Roopali Bhoite, Shahidul Islam, Guijun Yan, Md. Nurealam Siddiqui, Md. Omar Kayess, Kadambot H. M. Siddique

**Affiliations:** 1Department of Genetics and Plant Breeding, Sher-e-Bangla Agricultural University, Dhaka 1207, Bangladesh; 2The UWA Institute of Agriculture, The University of Western Australia, 35 Stirling Highway, Crawley, WA 6009, Australia; kadambot.siddique@uwa.edu.au; 3Grains Genetic Improvement, Department of Primary Industries and Regional Development, 3 Baron-Hay Ct., South Perth, WA 6151, Australia; roopali.bhoite@dpird.wa.gov.au; 4Department of Plant Sciences, North Dakota State University, Peltier Complex 2004A, 1300 18th Street, Fargo, ND 58108-6050, USA; shahidul.islam.1@ndsu.edu; 5UWA School of Agriculture and Environment, The University of Western Australia, 35 Stirling Highway, Crawley, WA 6009, Australia; guijun.yan@uwa.edu.au; 6Department of Biochemistry and Molecular Biology, Gazipur Agricultural University, Gazipur 1706, Bangladesh; nuralambmb@gau.edu.bd; 7Institute for Plant Sciences of Montpellier (IPSiM), University of Montpellier, INRAE, CNRS, Institute Agro, 34060 Montpellier, France; 8Institute of Biotechnology and Genetic Engineering (IBGE), Gazipur Agricultural University, Gazipur 1706, Bangladesh; kbdkayess@gmail.com

**Keywords:** wheat, cotton, oilseed brassica, climate resilient, disease, breeding, yield, quality

## Abstract

Polyploid crops such as wheat, Brassica, and cotton are critical in the global agricultural and economic system. However, their productivity is threatened increasingly by biotic stresses such as disease, and abiotic stresses such as heat, both exacerbated by climate change. Understanding the molecular basis of stress responses in these crops is crucial but remains challenging due to their complex genetic makeup—characterized by large sizes, multiple genomes, and limited annotation resources. Proteomics is a powerful approach to elucidate molecular mechanisms, enabling the identification of stress-responsive proteins; cellular localization; physiological, biochemical, and metabolic pathways; protein–protein interaction; and post-translational modifications. It also sheds light on the evolutionary consequences of genome duplication and hybridization. Breeders can improve stress tolerance and yield traits by characterizing the proteome of polyploid crops. Functional and subcellular proteomics, and identification and introgression of stress-responsive protein biomarkers, are promising for crop improvement. Nevertheless, several challenges remain, including inefficient protein extraction methods, limited organelle-specific data, insufficient protein annotations, low proteoform coverage, reproducibility, and a lack of target-specific antibodies. This review explores the genomic complexity of three key allopolyploid crops (wheat, oilseed Brassica, and cotton), summarizes recent proteomic insights into heat stress and pathogen response, and discusses current challenges and future directions for advancing proteomics in polyploid crop improvement through proteomics.

## 1. Introduction

Polyploid crops—with multiple sets of chromosomes—play an important role in global agriculture. They are the key drivers of crop evolution and contribute to global food security [1]. Polyploids exhibit greater vigor and yield potential than their diploid relatives. Whether natural or induced, polyploidization enhances heterozygosity, reduces inbreeding depression and mitigates deleterious mutations. Polyploids are classified into two major types: autoploids, with duplicated identical genomes [2], and alloploids formed by hybridization of more than two copies of divergent genomes [3]. Potato (*Solanum tuberosum*; 2n = 4x = 48 [4]) and sugarcane (*Saccharum* spp.; 2n = 62, 80, 96, 112, or 128 [5]) are the examples of autoploid crops, while wheat (*Triticum aestivum*; AABBDD; 2n = 6x = 42 [6]), Brassica species (*Brassica* spp.; 2n = 2x = 34 (BBCC), 36 (AABB) and 38 (AACC) [7]), and cotton (*Gossypium hirsitum*; AADD; 2n = 2x = 52 [8]) are allopolyploids. Other notable alloploids include oat (*Avena sativa*; AACCDD; 2n = 6x = 42), different millets (e.g., finger millet, porso millete, fonio millet, and Indian and Japanese barnyard millet), and quinoa (*Chenopodium quinoa*; AABB; 2n = 4x = 36) [8]. These polyploids contribute to our diet, industry, and economy. This review focuses on three globally significant allopolyploid crops—wheat, oilseed Brassica, and cotton.

Wheat (*Triticum aestivum*) is the most widely grown cereal crop, critical for food, animal feed and trade. In 2023–2024, global wheat production was estimated at 788 million tons (MT), with 209.6 MT traded [9]. However, production has declined while demand continues to rise, highlighting the urgent need for yield improvement.

Oilseed Brassica, a member of the Brassicaceae family, includes 4000 species, many of which are cultivated for edible oil [10]. Among them, *B. napus* (rapeseed/canola), *B. campestris*, and *B. juncea* are the most important [11]. However, *B. napus* is widely grown globally due to its low erucic acid and glucosinolate levels and high oleic and linolenic acid content. In contrast, *B. campestris* and *B. juncea*, which have higher erucic acid, are more regionally cultivated—*B. campestris* in temperate countries such as Australia, Germany, Sweden, France, and Canada, and *B. juncea* in Asia [12]. Global canola production reached 89.39 MT in 2023–2024, with a projected decline for 2024–2025 [13].

Cotton has industrial value in textiles, paper, oil, chipboard and livestock feed industries [14]. It has a value of around USD 600 billion in the global textile industry economy [15], with an estimated production of 24.12 MT in 2023–2024 [16].

The demand for these crops is rising due to global population growth—projected to increase by 25% by 2050 [17]. Wheat production must double by 2050 [18,19], cotton production must reach 28.1 MT by 2032 [20], and canola production in Canada is projected to increase by 3 MT by 2030 [21]. However, achieving these targets is hampered by biotic (e.g., pathogens and pests) and abiotic (e.g., heat, drought and salinity) stresses. These stresses disrupt plant phenotypes, physiology, biochemical, and metabolism, ultimately reducing yield and product quality. Among abiotic stresses, climate change–induced heat stress (HS) has become a major constraint to crop production, often causes drought by increasing evapotranspiration [22]. A 1 °C increase above 15 °C can reduce global wheat yields by 6% [23], and a similar increase above 30 °C can reduce canola and cotton yields by 7% [24] and 6.1% [25], respectively. An individual HS and combined HS and drought stresses reduce canola yield and oil concentration significantly [26], and impair cotton ball development [27]. In wheat, HS at the reproductive stage can cause pollen malformation and male sterility, [28,29,30] and a combined HS and drought stresses can cause mitotic and meiotic arrest, ovule abortion, and reduce stigma receptivity [31], drastically reducing grain number and quality. Besides abiotic stresses, biotic stresses are equally harmful. The presence of powdery mildew in wheat causes up to 33% yield loss [32], and clubroot disease in canola [33] and cotton leafroll dwarf virus [34] can result in up to 100% yield loss in severe cases.

Understanding the molecular basis of stress responses is essential for combatting these threats. During stress, the plant nucleus receives signals to alter gene expressions—exportation of mRNA from nucleus to cytoplasm and biosynthesis of stress-induced novel proteins [35,36]. To acclimatize to stress, the plant phenotype changes due to the biological function of the novel proteins. As biotic and abiotic stress responses, the biological function of a protein is governed by proteoforms—the basic molecular forms of proteins produced from a single gene including genetic variants, alternative splice variants, and post-translational modifications (PTMs) [37]—their cellular localization, and protein–protein and protein–other non-protein compound’s interactions [35,36]. Therefore, while genomics, transcriptomics, and phenomics have been useful to study changes in DNA, RNA [38], and phenotypes [39], respectively, proteomics provides more direct insights into stress adaptation mechanisms [40], while transcriptomics may not correlate with actual protein levels [41]. For example, transcriptomics techniques such as microarray and RNA-seq cover 48–57% of phenotypic variation and have moderate correlation with mRNA and protein (identified by label-free proteomics) expression [42]. Proteomics can identify the proteins and proteoforms actively involved in stress-related physiological, biochemical, and metabolic processes [43], and protein–protein interactions [44]. Proteomics is also a valuable tool for understanding the consequences of polyploidization [45].

Several proteomic tools are commonly used in polyploid crops to obtain stress-responsive molecular insights (Figure 1). Mass spectrometry (MS) has been the most extensively used tool in proteomics since its discovery in 1913 [46]. Techniques such as two-dimensional polyacrylamide gel electrophoresis–MS (2-DE-PAGE/MS) and its improved version, 2D-DIGE (2-dimensional difference gel electrophoresis), have evolved into advanced methods such as quadrupole (Q), time-of-flight (TOF), orbitrap, matrix-assisted laser desorption/ionization couple with TOF (MALDI-TOF), and shotgun proteomics [40,45]. Shotgun methods such as the label-based method, isobaric tags for relative and absolute quantitation (iTRAQ), and label-free quantitative proteomics are widely used. While label-free approaches offer precise quantification of all peptides of a sample, they are data-dependent acquisition (DDA) methods and therefore often fail to identify extensive low-abundant proteins or peptides [47]. To overcome this challenge, sequential window acquisition of all theoretical fragment ion spectra mass spectrometry (SWATH-MS), a data-independent acquisition (DIA) method, is being applied in crops-, including wheat [48,49] and cotton [50].

**Figure 1 proteomes-13-00060-f001:**
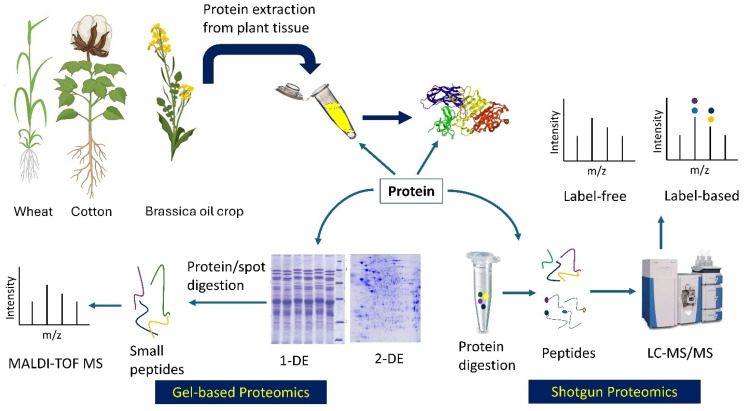
Common proteomic approaches used in allopolyploid crops: Gel-based (**bottom left**) and shotgun (**bottom right**). The gel photos were modified from Gris and Baldoni (2013) [51] and Nadeem et al. (2016) [52]. Crops were generated from BioRender (https://BioRender.com/, accessed on 3 October 2025). Stable isotope labels for label-based method were represented by different colors; 1-DE and 2-DE = one- and two-dimensional gel electrophoresis; MALDI-TOF = matrix-assisted laser desorption ionization–time-of-flight; MS = mass spectrophotometry; LC = liquid chromatography.

Despite these advances, proteomic research in polyploid crops remains challenging due to genome complexity, inefficient protein extraction methods, lack of organelle-specific databases, insufficient protein annotations, and limited protein validation [40,53,54,55]. The genomic complexity of allopolyploids arises from the genomic shocks due to hybridization causing genome rearrangement and altered gene expression compared to their diploid parents [56]. Due to multiple genomic interactions between progenitor genomes, chromosomal rearrangements, genetic and well as epigenetic modifications, and subgenome biasness occur, which lead to significant differences in their gene, protein, and metabolite expression [57,58]. Additionally, cytonuclear co-evolution commonly occurs among polyploids, leading to cytonuclear incompatibility causing gene expression and protein product differentiation between the nucleus and cytoplasm [59,60]. However, genomic plasticity allows the allopolyploids to adapt with these significant changes—alteration in chromosome number and structure (insertion, deletion, and translocation), genome sizes, and epigenetics—and reconstruct the genes and proteins for different phenotypes for their adaptability [58]. New phenotypes of the polyploids compared to their parents are the result of the genomic complexity of their transcriptome, proteome, and metabolome, while orthologous protein interactions determine the parental gene expression in the allopolyploids [61]. Therefore, genome sequence or transcriptome analyses are insufficient to understand the adaptability mechanisms of polyploids due to polyploidization and environmental changes [62,63]. Compared to diploids, the proteomic data interpretation in the polyploids can be challenged by “homeologs collapse”—the wrong assembly of two homeologs into a single gene [64]—and “homeologe expression bias”—the unequal expressions between homeologs caused by the duplicated gene actions [65]. For the first time, identification of “homeologe expression bias” in the protein data between the A and B subgenomes in *Arachis hypogaea* provides insights into the biological processes involved in homeologous and paralogous protein expression [66].

However, recent advancements in genome sequencing, functional and subcellular proteomics, improved extraction methods, and the identification of stress-related protein markers highlight the growing potential of proteomics for enhancing stress resistance in polyploid crops. Though proteomics identifies the gene response for environmental stimuli, multi-omics—the integration of other omics such as genomics, transcriptomics, epigenomics and metabolomics with proteomics—could provide a more comprehensive understanding of stress tolerance in polyploids than the single approach [67].

This review highlights the genomic complexity of polyploid crops—wheat, oilseed *Brassica*, and cotton—and the progress of their proteomics for HS and biotic stress (disease), focusing on comparative proteomics, challenges of proteomics in identifying functional proteins and proteoforms of polyploids, and the prospects of proteomics, particularly subcellular proteomics, in polyploid breeding.

## 2. Genome Complexity of Polyploidy

### 2.1. Wheat

Bread wheat (*T. aestivum*; AABBDD; 2n = 6x = 42), a hexaploidy species in the Poaceae family, is one of the world’s most important staple cereal crops. It evolved naturally through two successive hybridization events. First, a diploid wheat (*T. urartu*; AA; 2n = 2x = 14) crossed with a wild goat grass (*Aegilops speltoides*; BB; 2n = 2x = 14), producing a tetraploid wild emmer wheat (*T. turgidum* spp. *diccoides*; AABB; 2n = 4x = 28). This tetraploid then hybridized with another goat grass (*A. tauschii*; DD; 2n = 2x = 14), resulting in modern hexaploidy bread wheat (Figure 2a) [6,68]. The estimated genome sizes of hexaploid, tetraploid, and diploid wheat are approximately 17 Gb [69], 10.1 Gb [70], and 4.94 Gb [71], respectively.

### 2.2. Oilseed Brassica Crops

The genus *Brassica*, within the Brassicaceae family, includes economically important species cultivated for oilseeds, vegetables, and condiments [72]. The *Brassica* genus has three diploid species—*B. nigra* (BB; 2n = 16), *B. oleracea* (CC; 2*n* = 18), and *B. rapa* (AA; 2n = 20). They evolved, by intercrossing into three amphidiploid species: *B. carinata* (BBCC; 2n = 34), *B. juncea* (AABB; 2n = 36), and *B. napus* (AACC; 2n = 38) (Figure 2b) [7]. Among the three diploids, *B. rapa* is the oldest and most widely distributed [73], with a genome size ranging from ~389.2 to 424.59  Mb [74,75]. The genome sizes of *B. oleracea* and *B. nigra* are estimated at ~587.7–630.7 Mb [76,77] and ~446.5–534.2 Mb [78,79], respectively. The genome sizes of the amhidiploids are approximately 922–961.72 Mb for *B. juncea* [80,81] and 921.5–975.8 Mb for *B. carinata* [82].

**Figure 2 proteomes-13-00060-f002:**
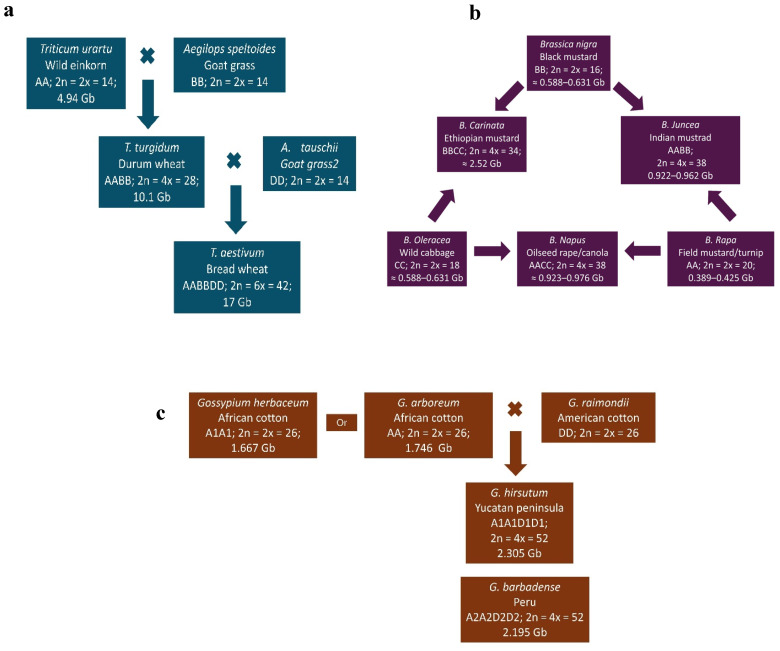
Evolution and genome structure of three major allopolyploid crops: (**a**) wheat (*Triticum aestivum* L.), (**b**) oilseed *Brassica* crops, and (**c**) cotton (*Gossypium hirsutum* and *G. barbadense*); adapted from [83].

### 2.3. Cotton

Cotton, belonging to the genus *Gossypium* and Malvaceae family, has been cultivated for more than 7000 years [84] for fibers, medicine, and animal feed [85]. Although there are around 50 species, only four are widely cultivated: two American species—*G. hirsutum* (A1A1D1D1; 2n  =  4x  =  52) and *G. barbadense* (A2A2D2D2; 2n  =  4x  = 52)—and two Asian and African species—*G. arboreum* (AA; 2n  =  2x =  26) and *G. herbaceum* (A1A1; 2n  =  2x  =  26) (Figure 2c) [83,85,86,87]. Forty-three *Gossypium* species are diploid (n = 13), classified into genome groups A–G and K, while seven are tetraploid (AADD) [88,89]. The A-genome originated in Africa and the D-genome in Mexico [88]. Allotetraploid or amphidiploid cottons species—*G. hirsutum* (2305.24 Mbp), *G. barbadense* (2195.8 Mbp), *G. tomentosum* (2193.56 Mbp), *G. mustelinum* (2315.09 Mbp), *G. darwinii* (2182.96 Mbp) [90], *G. ekmanianum* (2341.87 Mbp), and *G. stephensii* (2291.84 Mbp) [91]—are the result of polyploidization between *G. arboreum* (A genome; 1746 Mb; [92] or *G. herbaceum* (1667 Mb; [93]) and *G. raimondii* (D genome; 885 Mb; [94,95,96]). Among these cotton species, *G. hirsutum* is the most widely cultivated species globally [97].

## 3. Proteomic Studies on Heat Stress Tolerance in Polyploid Crops

Heat stress significantly impairs crop growth, development, yield and grain, and seed, or fiber quality. Although HS can adversely affect all of a plant’s developmental stages (germination, seedling, vegetative and reproductive phases, and maturity), the anthesis and post-anthesis phases are particularly sensitive in crops like wheat, oilseed brassica, and cotton. Proteins—functional gene expression products—play key roles in a plant’s HS response by regulating various metabolic pathways. For example, HS increases biomolecule kinetics, resulting in protein degradation or misfolding risk; protective proteins or proteoforms accumulate to reduce the harmful effect of HS on the cells [35]. Different HS-induced proteins, particularly heat shock proteins (HSPs), and proteoforms are strongly associated with HS tolerance across different developmental stages. This section discusses how HS affects crops, and the molecular mechanisms of the HS response revealed by proteomics approaches.

### 3.1. Wheat

In wheat, HS adversely impacts every stage, from seedling emergence to maturity, ultimately reducing grain yield and quality [98]. Heat stress during tillering disrupts pollen viability, spike formation, seed set, and embryo development, while terminal HS (at reproductive stage) is detrimental to yield [99]. Post-anthesis HS inhibits photosynthates translocation and starch accumulation into grains and reduces of carbohydrate content in the stem. It also leads to increased reactive oxygen species (ROS), osmolyte accumulation, decreased RuBisCo activity, and degradation of seed proteins [100]. However, the accumulation of HSPs, antioxidant enzymes (e.g., catalase (CAT), peroxidase (H_2_O_2_), superoxide dismutase (SOD) and glutathione (GSH)) and proline, along with the suppression of anabolic and catabolic processes, are critical for HS tolerance [40,99]. Numerous proteomic studies in recent years have examined HS responses at different growth stages in wheat (Table 1). Our previous review summarized recent progress in wheat proteomics under HS [40].

Proteomic studies identified that proteins associated with photosynthesis and PTMs play important roles in HS tolerance in wheat seedlings. Cytoplasmic proteins such as 2-cysteine peroxiredoxin (*Ta2CP*) in wheat contribute to HS tolerance and chlorophyl metabolism in seedling stage [117]. *Arabidopsis* with overexpressed *Ta2CP* had normal green leaves with high chlorophyll a and b content and low H_2_O_2_ and ROS, leading to thermotolerance. The application of 2-DE and MALDI TOF/TOF proteomics identified that the upregulation of photosynthesis-associated proteins, including RuBisCO activase A and PEP carboxylase, and the signal transduction pathway in wheat contributes to HS tolerance by increasing CO_2_ assimilation [118]. A small HSP gene, *TaHSP23.9* (a member of the HSP20/ACD family), was significantly upregulated in wheat leaves and developed grain under HS at both the transcription (mRNA) and translation levels, making it a promising marker for HS tolerance [106]. Phosphorylation, a major PTM, also plays an important role in HS tolerance. Due to the amino acid substitution in the *TaSG-D1* allele, the *TaSG-D1^E286K^* gene undergoes phosphorylation and stabilizes the expression of *TraesCS5B02G380200* (TaPIF4), resulting in improved HS tolerance in the seedling and mature stages of *T. sphaerococcum* [119]. In rice, phosphoproteins contribute to heat-stress signaling by decreasing RuBisCo and ATP synthesis, stabilization of HS transcripts, and transduction of cellular-signaling due to dephosphorylation [120].

Proteins associated with photosynthesis and male sterility improve grain yield and quality in wheat. Sowing time affects photosynthesis and plays a crucial role in mitigating HS effects. For example, delayed sowing lead to higher photosynthetic rates in flag leaves, which resulted in 11.16% and 10.10% increases in grain weight and yield in wheat, respectively [102]. This improvement was linked to increased energy flow into the electron transport system and the upregulation of differentially abundant proteins (DAPs) involved in photosynthetic electron transport (e.g., PsbH and PsbR), the Calvin cycle, and chlorophyll biosynthesis. Conversely, terminal HS under standard sowing dates significantly reduces photosynthesis and starch accumulation, diminishing grain yield and quality, respectively. Combined 2D-PAGE and transcriptomic analyses revealed that mitogen-activated protein kinase (MAPK) contributes to terminal HS tolerance in wheat by promoting proline, starch, and H_2_O_2_ accumulation, enhancing granule integrity, positively regulating stress-related gene expression and amylolytic activities, while suppressing photosynthesis-related genes [104]. Moreover, MAPK improves oxidative stress tolerance through its positive correlations with SOD and HSP17. The role of MAPK in HS tolerance in other crops including rice, corn, cotton, potato, and tomato has been reviewed earlier [121]. MAPK signaling is also involved in HS-induced male sterility in wheat. The downregulation of MAPK-related genes (e.g., *TraesCS3A02G149800.1*) under HS negatively affects ROS scavenging and pollen development [28]. Furthermore, DAPs related to phenylpropanoid biosynthesis accumulates excess ROS in anthers and inhibit starch and sucrose synthesis through phenylpropanoid biosynthesis, resulting in anther indehiscence. Upregulation of the HS pathway thermosensitive male sterile 1 gene, *TraesCS4B02G193500.2*, causes male sterility under HS.

ABA biosynthesis and proline accumulation improve HS tolerance in the spike differentiation stage in wheat [48]. A comparative transcriptomic and proteomic study across three spike differentiation stages identified 210 HS-associated transcripts/proteins. Downregulation of 9-cis-epoxy carotenoid dioxygenases and upregulation of zeaxanthin epoxidase and ascorbate oxidation modulate ABA biosynthesis. Similarly, ornithine aminotransferase and pyrroline-5-carboxylate reductase enhance proline accumulation via ornithine and glutamate pathways. Phosphorylation regulates reproductive organ development and photosynthetic apparatus in leaves in wheat under HS [122]. A DDA LC-MS/MS identified 14 phosphoproteins under HS, which activated phosphatases and kinases and generated their phosphoforms. A protein kinase, TraesCS6B01G377500.3, was associated with upregulated S711 and S762 phosphosites. Serine or threonine dominated phosphosites—S3236 and T3238—maintained protein functions by preserving the phosphorylation pool under HS. However, identification of proteins and proteoforms is recommended to understand the co-regulation of these phosphosites to combat HS.

Heat stress affects wheat grain weight and quality. A 2-DE and MALDI-TOF showed that HS reduced grain weight by decreasing starch synthesis proteins such as glucose-1-phosphate adenyl transferase, although catalase protected cells from H_2_O_2_ [123]. Heat stress also weakens dough properties due to reductions in glutenin and gliadins, and an increase in HSPs, although sHSPs improve thermotolerance during grain filling stage [124].

Increased proline and H_2_O_2_ levels in flag leaves leads to 9% more grain weight lost under combined day–night temperature HS conditions than high daytime temperatures alone [103]. Day–night temperatures reduce thermotolerance, grain yield, and quality by restricting the expression of transcription factors (TFs), genes associated with signaling pathway and abiotic stress, and the function of protein folding machinery. Using 2D-PAGE and MALDI-TOF, 153 and 95 DAPs were identified in wheat grain under day and day–night HS conditions, respectively. Proteins associated with starch biosynthesis, cell wall synthesis, defense, and storage were altered, with the particularly noteworthy downregulation of starch and cell wall-related proteins, resulting in poor grain development and premature senescence in both conditions.

Heat stress also affects flour quality and dough quality by altering protein abundance in wheat grain [108] and by modulating the abundance of storage proteins [99], respectively. A decrease in starch accumulation and an increase in gluten, gliadin, globulin, and albumin content was shown to result in a reduced thousand grain-yield in wheat [108]. The downregulation of HSP90, low-molecular-weight glutenin subunits (LMW-GSs), and starch branching enzyme IIb reduced both grain yield and quality. Five DAPs—elicitor-responsive gene 3, brassinosteroid-insensitive 1, histone cell cycle regulator, chaperone protein, and splicing arginine serine-rich 7—were also linked to protein–protein interactions affecting grain yield and quality. Heat stress affects dough quality in wheat grain by affecting high-molecular-weight glutenin subunits, LMW-GSs, α- and γ-gliadins, and avenin-like proteins [101].

Moreover, peroxidase, RuBisCo activase, sHSPs, and photosynthetic electron transport are crucial for thermotolerance during the seedling stage, while proteins related to antioxidants, MAPK, proline and H_2_O_2_ accumulation, and phosphofroms are vital during the grain filling stage. Globulin, gliadin, and albumin are essential in determining grain quality under HS.

### 3.2. Oilseed Brassica Crops

Heat stress affects all growth stages of the oilseed *Brassica*, but the flowering stage is the most sensitive [125]. It reduces gametophyte fertility, pollen viability, germination, and causes seedless pod resulting in a decreased thousand grain weight, reduced yield, and poor oil quality [125,126]. Among the cultivated species, *B. napus* is more sensitive to HS than *B. juncea* and *B. campestris* [126]. Proteins such as HSPs (e.g., HSP101, HSP70 and HSP17.6) [125], gucosinolates, including glucobrassican, glucocarphanin and glucoraphasatin, gluconapin, sinigrin, and progoitrin [127] and glyoxalase I [128] contribute to thermotolerance. Phytohormone signaling pathways also play a critical role in HS-tolerance.

Proteomic study of the effect of HS on *Brassica* started only a decade ago. The first proteomic study on HS in *B. napus* found ascorbate peroxidase as a key protein for thermotolerance in seedlings using 2-DE [129]. Increased glyoxalase-I enhanced seed thermotolerance by detoxifying methylglyoxal, maintaining redox balance, and increasing SOD activity [128].

Beyond canola, proteomic studies have also been conducted on other oilseed *Brassica* species. Over the last five years, the advancement of proteomic techniques has accelerated in HS responses in oilseed *Brassica* (Table 1). Heat stress causes dead organs such as seed coat and pericarp affecting yield [130], while label-free proteomics identified the role of protease accumulation in dead pericarps in *B. juncea* [131]. A combined proteomic and metabolomic study on *B. juncea* under HS found the high abundance of several proteins, including HSP70-5, HSP104, and multiple sHSPs, with proteases being the most abundant in the pericarp [131]. Heat primarily induced the accumulation of cysteine proteases, 25 HS-induced proteins, and sugars on its dead pericarp.

Increased antioxidants and flavonoids contribute to thermotolerance in *B. juncea* seedling [114]. Phenylalanine ammonia lyase and respiratory burst oxidase homolog protein-A improve thermotolerance by synthesizing metabolites (e.g., flavonoids and lignins) and producing ROS, respectively. A combined proteomics and peptidomics—a comprehensive analysis of all peptides of an organism [132])—study in oilseed *Brassica* identified thermotolerant DAPs. Key thermotolerance-associated proteins, such as HSPs (21.7, 14.7 and 17.5 kDa HSP groups), sHSPs, and TFs, including HS transcription factor A-4a, Myb family Atlg14600, and WRKY, are related to signal transductions, regulating transcriptions, repairing DNA, and processing RNA. In *B. campestris* seedling, thermotolerance under both heat and cold stress involves several molecular pathways, including redox homeostasis, photosynthesis, chaperons, HSPs, carbohydrate metabolism, and signal transduction [116]. Under both stresses, 1022 DAPs (172 upregulated and 324 downregulated) and 1784 DAPs under HS were found using a combined TMT and LC-MS. The critical role of redox homeostasis in thermotolerance was validated through the reduced glutathione-to-oxidized glutathione ratio and decreased oxidative damage in transgenic *Arabidopsis* overexpressing *GLU1*, a gene encoding a redox-associated protein.

### 3.3. Cotton

Heat stress critically affects all the developmental stages of cotton, with the reproductive stage being the most sensitive [133]. It impairs seed germination, shoot and root growth, and induces male sterility, ultimately reducing yield and fiber quality. Increased leaf temperature due to HS disrupts photosynthesis by inhibiting RuBisCO activation [134]. In *G. barbadense*, HS restricts photosynthesis by limiting the regeneration capacity of ribulose-1,5-bisphosphate and impeding electron transport [135]. Traits such as membrane thermostability, high proline and soluble sugar can improve thermotolerance in cotton [136]. Additionally, the regulation of microRNAs, signaling pathways, including calcium, ROS, carbohydrate, transcription factors, phytohormones and gene regulation pathway, and HSPs play critical role in its thermotolerance [133]. Despite numerous studies on the physiological impacts of HS on cotton, proteomic studies remain limited. However, recent proteomic studies have provided valuable insights into thermotolerance in cotton (Table 1).

Heat stress increases ROS and mitochondrial acetylation activity and decreases ATP levels in the cotton seedling’s leaves leading to the HS-sensitivity through the expression of the mitochondrial protein, GhHSP24.7 [111]. However, suppressed GhHSP24.7 enhanced thermotolerance by decreasing stomatal conductance and H_2_O_2_, activating GhHDA14, maintaining chloroplast structure, and improving ROS scavenging.

The reproductive stage of cotton is particularly vulnerable to HS [137]. Upregulated protective proteins such as HSP (15.7 kDa) and peptidylprolyl isomerase recovers pollen viability at the tetrad stage under 38 °C [50]. However, under 40 °C, at the tetrad stage, mature pollen grain size and pollen tube are reduced by 36% and 120 µm, respectively, resulting in a yield reduction of two-thirds [111]. Protein export and folding, and their processing pathways protect pollen from HS through upregulation of HSP70, isoforms of heat shock cognate 70 kDa protein 2-like, luminal-binding protein 5-like, and luminal-binding protein 5-like, HSP70 protein 14–15-like and HSP70-90 organizing protein 3-like and downregulation of late embryogenesis abundant protein (LEA) 2.

A recent proteomic study identified over 8,000 HS-induced DAPs at the flowering stage [109]. Increased abundance of β-glucosidase, different photosynthetic proteins, NDH subunit of subcomplex B1, NADH ubiquinone oxidoreductase (Complex I), stress-responsive proteins, 1-aminocyclopropane-1-carboxylate oxidase, and HSPs (e.g., Hsp70, Hsp90 and Hsp80) is crucial for thermotolerance. These proteins contribute to HS-tolerance by maintaining membrane and cell wall integrity, producing energy, maintaining osmotic balance, enhancing proline accumulation and ROS scavenging, and regulating electron transport in photosynthesis channels, carbohydrate metabolism, and protein folding, and preventing protein aggregation. The role of HSP70 has also been reported for HS and drought tolerance in rice through increasing proline accumulation and SOD activities, respectively [138].

Heat stress also reduces insecticidal proteins in *Bt* cotton’s bolls [112]. Under combined heat (38 °C) and drought (40% field capacity) stress, the insecticidal proteins led to a decrease in the fresh weight of boll shell by 38.3 and 30.7 ng/g in 2026 and 2017, respectively. That reduction may be the result of the downregulation of signal recognition particles (SRPs) and SRP receptors, restriction of peptide chain translocation into the endoplasmic reticulum, and increase in ubiquitin-mediated proteolysis.

Overall, proteins associated with photosynthesis, carbohydrate metabolism, stress response, ROS regulation, protein folding and aggregation, and electron transport play pivotal roles in cotton growth and HS tolerance.

## 4. Proteomic Studies for Biotic Stress

Biotic stress—caused by living organisms such as bacteria, fungi, viruses, and insects—significantly threatens global food security [139]. Plant resistance to pathogens is a complex trait governed by a multifaceted interplay of morphological, genetic, biochemical, and molecular processes, with proteomic insights playing pivotal roles [140,141,142]. Therefore, a comprehensive understanding of these mechanisms is essential in order to address biotic stress in complex plant genomes. Proteins and PTMs contribute directly to plant immunity through activating stress-associated defense responses, while subcellular protein localization provides key insights into how plants respond to biotic stress. Therefore, proteomics—a powerful and evolving analytical tool—offers substantial potential for developing disease resistance, particularly in polyploid crops [143,144].

### 4.1. Wheat

Wheat production is mostly challenged by fungal pathogens such as rusts, mildew, and blights [145]. Numerous proteomic studies have explored the mechanisms underlying wheat–pathogen interactions to understand both compatible and incompatible responses [146,147,148,149,150]. For example, a comprehensive proteomic revealed key virulence and host defense proteins—including PR1, PR4, and GST—associated with wheat rust resistance [150]. In the wheat apoplast, an increased level of protein-modifying enzymes, particularly serine proteases, mitigated infection-induced damage by *Magnaporthe oryzae* [151]. Upregulation of antioxidant-related proteins and downregulation of plant–pathogen interactions and glutathione metabolism were found due to *Tilletia controversa* and *T. foetida* infection in wheat [152]. However, thaumatin-like proteins and HSPs contributed to increasing the resistance against these pathogens.

Separately, 366 DAPs for wheat infected by *Fusarium pseudograminearum* (causal agent of Fusarium crown rot) were identified using the TMT tool [153]. Cellular activities—glucoside, electron transport, cellulose synthase and oxidoreductase—metabolic processes, plant–pathogen interactions, and antioxidant activities contribute to resistance to the disease [153,154]. In the resistant variety Xinong 538, unique proteins encoded by *Chitinase IV*, *Thaumatin-like 1*, *PR1.1*, and *PR1.2* genes were strongly associated with resistance to Fusarium head blight (FHB) [154]. Furthermore, an FHB resistant network—Ca^2+^, phytohormone- and phenylalanine-signaling pathways for stomatal closure, disease resistance and thickening cell wall, respectively—was proposed. Although wheat cultivars vary in FHB susceptibility, they share a core set of defense-related proteins [155]. During FHB infection, pectin-derived oligogalacturonides (produced via homogalacturonan degradation) act as damage-associated molecular patterns. They are recognized by wall-associated kinase 1 in *Arabidopsis*, a plant receptor kinase functioning as a pattern recognition receptor. Similarly, the *Stb6* gene in wheat encodes a wall-associated kinase-like protein that recognizes *AvrStb6*, the fungal avirulence effector, thereby conferring resistance to *Septoria tritici* blotch [146]. In the resilient wheat cultivar “Xingmin318”, 741 DAPs had distinct defense strategies against *Puccinia striiformis* and *Blumeria graminis* infections [156], while 394 DAPs showed resistance in response to *Blumeria graminis* f. sp. *tritici* infection [157]. Among these, a few DAPs were primarily associated with pathogenesis-related processes, oxidative stress responses, and primary metabolism. Key metabolic pathways for plant’s defense response include phenylpropanoid biosynthesis, phenylalanine metabolism, and photosynthesis-antenna proteins. The identification of protein motifs rich in leucine repeats and histidine sites, along with eight predicted PPI networks, provides insights into the potential molecular mechanisms underlying resistance to powdery mildew.

Although proteomics has significantly advanced our understanding of plant–pathogen interactions at the molecular level, the narrow genetic diversity of modern wheat varieties remains a major challenge. Continuous research and breeding programs are essential to enhance wheat’s resistance to evolving pathogens.

In summary, proteins involved in antioxidant defense, glutathione metabolism, carbohydrate metabolism, photosynthesis, and calcium-, phytohormone-, and phenylalanine-mediated signaling pathways play vital roles in wheat’s resistance to biotic stress.

### 4.2. Oilseed Brassica Crops

Biotic stress activates detoxification, redox homeostasis, and defense-related pathways in oilseed Brassica. Black rot, caused by *Xanthomonas campestris* pv. *campestris* (*Xcc*), severely affects cruciferous vegetables, including *Brassica napus* [158]. Five major functional protein groups—antioxidative systems, proteolysis, photosynthesis, redox, and innate immunity—were identified in *Xcc*-responsive DAPs in the susceptible cultivar (Mosa) and the resistant cultivar (Capitol) using label-free proteomics. Mosa showed increased protein degradation and oxidative stress markers, while Capitol exhibited high redox potential and an enhanced innate immune response. Downregulation of photosystem II-related proteins improved disease resistance.

Using LC with tandem MS, 73 putative proteins were identified on the quantitative trait loci in chromosomes A3 and A8 in the clubroot (caused by *Plasmodiophora brassicae*) resistant *B. napus* cultivar [159]. The abundance of the proteins associated with pathways related to calcium signaling, reactive oxygen species, dehydrins, lignin, and phytohormones changes due to the pathogenic infestation. Additionally, 73 orthologous proteins, including calcium signaling associated proteins (BnaA02T0335400WE, BnaA03T0374600WE, BnaA03T0262200WE, and BnaA03T0464700WE) play a crucial role in defense against *Plasmodiophora brassicae*. In *B. oleracea*, proteins associated with ABA and glucose signaling, and fructose-bisphosphate aldolase are critical for clubroot defense [160]. Most of the abundant proteins were localized to outside the nucleus, particularly in the chloroplast, thylakoid, stroma, and membranes. Additionally, the restriction of energy metabolism enhances resistance.

Stem rot, caused by *Sclerotinia sclerotiorum*, is another major disease of *Brassica* [161]. In the first proteomic study on *S. sclerotiorum* infection in *B. napus*, upregulated antioxidant enzymes peroxidase and SOD along with proteins related to photosynthesis, metabolism, hormone signaling, and protein folding were reported for the disease resistance [162]. Several proteomic studies have expanded the understanding of stem rot resistance mechanisms in *B. napus* [161,163,164,165]. A TMT-based analysis identified 221 unique proteins and 173 DAPs in response to stem rot infection, including proteins involved in ROS homeostasis, lipid signaling, DNA methylation, histone modification, defense responses, and cyanate lyase activity [165].

In *B. juncea*, nano-LC-MS revealed peptidyl-prolyl cis–trans isomerase as the key protein for stem rot defense [166]. Furthermore, interacting with other proteins related to protein importation into the nucleus and RNA exportation, including GTP binding nuclear proteins, 60S ribosomal protein and protein kinase play important roles in disease defense. Alternative splicing on introns or alternative 3′ splice sites in *S. sclerotiorum*-infected *B. napus* led to 130 genes with 98 differentially expressed genes for defense [167]. However, further proteomic investigations focusing on PTMs are needed to deepen our understanding of host–pathogen interactions.

Overall, proteins associated with photosynthesis, ROS homeostasis, lignin synthesis, lipid- and Ca^2+^- signaling, DNA methylation, and histone modification contribute to the biotic stress resistance in oilseed Brassica.

### 4.3. Cotton

Phytopathogens—fungi, bacteria, and viruses—cause significant yield loss in cotton [168]. Among the fungi, *Verticilium dahlia*, *Fusarium oxysporum*, *Thielaviopsis basicola* [169], and *Rhizoctonia solani* [170] are major causal agents for cotton diseases. Bacterial blight by *Xanthomonas campestris* also causes significant yield loss in cotton [171]. Cotton proteomics has advanced significantly over the past decade, improving the understanding of pathogenicity and plant defense responses at the molecular level [53].

Numerous proteomic studies have focused on unraveling the molecular mechanism of Verticillium wilt (VW), a devastating soil-borne disease caused by *V. dahlia* that severely affects cotton yield and quality [172]. The first root proteomic study on VW in cotton (*G. barbadense*) identified 51 upregulated and 17 downregulated proteins using 2-DE [173]. This study highlighted the roles of the ethylene signaling pathway, pentose phosphate pathway, and Bet v 1 family proteins in defense. The proteins of Bet v 1 family proteins, PR10 (GbPR10.5D1), increased resistance by modulating lipid metabolism and defense signaling pathways [174]. A DIA-based comparative proteomics identified 885 DAPs associated with disease resistance in *G. barbadense* [175]. Silencing ascorbate peroxidase proteins reduced VW resistance, underscoring the importance of APX-mediated redox metabolism in plant defense. Gene co-expression network analysis identified oxidoreductase and peroxidase activities as significantly enriched pathways for VW resistance.

Silencing a nuclear protein, *GthGAPC2*, in *Gossypium thurberi* increased susceptibility of VW by reducing lignin content and disrupting redox balance [176]. On the other hand, overexpression of *GthGAPC2* in tobacco and *Arabidopsis* confirmed its role in disease resistance, highlighting its potential as a promising candidate for breeding programs. Mi et al. (2024) [177] demonstrated that the GhMPK9-GhRAF39_1-GhWRKY40a signaling pathway enhanced cotton disease resistance by regulating the GhERF1b- and GhABF2-mediated pathways. *GhMAC3e*, a homologous gene of a key component of the MOS4-associated complex, was implicated in growth and defense responses against VW [178]. A phosphoproteomic study identified 359 and 287 differential phosphoproteins at 1 and 3 days post-VW inoculation in the resistant line, respectively [179]. Analysis of PTMs revealed 37 enriched serine (Ser) and 4 threonine (Thr) motifs, of which 19 Ser and 2 Thr were novel. Moreover, the upregulation of burst oxidase homolog D (*GhRbohD*) increased phosphorylation after VW infection, increasing resistance by decreasing ROS, H_2_O_2_, nitric oxide, and calcium levels while promoting lignin and cellulose accumulation [180].

Fusarium wilt (FW), caused by *F. oxysporum*, also results in considerable yield losses in cotton [181], though limited proteomic studies have explored this disease. Proteins associated with polyketides and cellular processes (e.g., lipid biosynthesis, protein and nucleotide metaboSerlism, protein folding, carbohydrate metabolism and oxidative stress response), HSP70, and protease are significantly enriched in the *F. oxysporum* f. sp. *vasinfectum* [182]. These proteins are crucial in mediating host–pathogen interactions via extracellular vesicles. In contrast, the MAPK cascade involving *GhMPK20*, *GhMKK4*, and *GhWRKY40* negatively regulated FW resistance, while *GhMPK20* plays a key role in the defense signal transduction pathway to improve FW resistance in cotton [183]. Furthermore, a phrosphoproteomics identified *GhMORG1* (a MAPK scaffold protein of the *GhMKK6*–*GhMPK4* cascade) as enhancing FW resistance in cotton [184].

In response to *Rhizoctonia solani* infection, iTRAQ analysis identified 174 DAPs, with proteins involved in ROS homeostasis—such as glutathione S-transferases and glutathione peroxidase—and those linked to lignin and phenylpropanoid biosynthesis, DNA methylation, and histone modification significantly upregulated [185].

Defense mechanisms for bacterial blight, caused by *Xanthomonas campestris* pv. *Malvacearum*, include the downregulation of photosynthesis-related proteins, such as ribulose-biphosphate carboxylase activase, ATP synthase β subunit, RuBisCo large subunit-binding protein, and glyceraldehyde-3-phosphate dehydrogenase B [186]. Furthermore, ROS-regulating proteins, including peroxidase, peroxiredoxins, and Prx type 2-Cys, play important roles in enhancing resistance.

Overall, proteins associated with signaling pathways—ethylene, calcium, and defense—as well as lipid and carbohydrate metabolism, DNA-methylation, histone modification, and peroxidase activity contribute to the biotic stress resistance in cotton.

## 5. Challenges of Proteomics in Polyploid Crops

Proteomics has been used in allopolyploids, including wheat, oilseed Brassica, and cotton, to harness the molecular insights of abiotic and biotic stress tolerance. However, these studies face several challenges, primarily due to the complex nature of polyploid genomes, difficulties in protein identification, reproducibility, and the lack of comprehensive protein databases.

The genomic complexity of an allopolyploid is the result of the combination of divergent genomes through interspecific hybridization. Gene loss during interspecific hybridization or further polyploidization (e.g., in wheat), continued recombination between the homoeologous chromosomes (e.g., Brassica), and chromosomal translocations between subgenomes increase the complexity of the genomic landscape [187,188]. While proteomics analysis can detect chromosomal rearrangements by comparing DAPs or proteoforms between progenitor lines, the specific parental contributions often remain unresolved [45].

For example, bread wheat has a large and complex genome (~17 Gb across three sets of chromosomes) [69], with 55–63% comprising repetitive sequences [189]. Paralogous genes in allopolyploids often share > 90% amino acid sequence identity. This high redundancy makes it difficult to identify novel proteins and protein–protein interactions associated with distinct phenotypes [45,190]. Studies have shown that 65–70% of transcriptomic data does not correspond to proteomic outcomes [41], potentially due to its amino acid composition and alternative splicing or PTMs. Protein isoforms—protein sequences altered by alternate splicing—and PTMs are two major proteoforms for determining biological functions of proteins in plants [36].

Besides genetically functional proteins, the identification of proteofroms is crucial due to their important role in allopolyploid evolution and stress tolerance. Homologous genes—orthologous and paralogous—and homeologous genes are a significant part of polyploid evolution. Homeologous genes or homeologs are the homologous gene pairs originated from different species but merged to an allopolyploid genome by alloplolyploidization [191], while protein isoforms are formed due to their expression [192]. For example, 18.7% of expressed genes of allotetraploid wheat (AADD) have a similar type of alternate splicing in two subgenomes [192], while 90 novel protein isoforms of a HSP, *TaHSP90*, developed from three homeologs were found in hexaploid wheat [193]. The protein isoform of TaYRG1.1 (TaYRG1.6) increases susceptibility to yellow rust disease in wheat [194], while *TaHSFA6e*, originated from the alternative splicing of a HS transcription factor gene increases HS-tolerance [195]. However, the identification of protein isoforms is challenging in allopolyploids due to a lack of complete protein databases, a higher chance of shared peptides compared to diploids due to their homeologous genes, and variability of the isoforms according to developmental stages of crops, tissue types, and external stimulus [196]. Long-read proteogenomic pipelines or prediction models using artificial intelligence (AI) can help overcome the challenges.

Similarly, PTMs also play an important role in stress tolerance in polyploids. Recently, the model of controlling cellular functions by PTM was proposed as follows; enzymes such as kinases, “writer”, catalyze the PTM addition to amino acid residues, while other enzymes such as phosphatases de-ubiquitinases and -acetylases, “erasers”, play the role of deleting PTMs from target amino acid residues, while a protein domains, “readers”, play the role of interacting between the PTMs and the other proteins [197]. However, this PTM crosstalk is underexplored in plant proteomes, which can be addressed by incorporating AI. The use of AI in identifying critical PTMs has been significantly advanced in human proteomics. There are many AI-based models, including machine learning and deep learning models, that have been developed to predict sequence for PTM sites, as well as their structure and their functions [198]. For example, a protein language-based model, PTMGPT2, precisely identified 19 disease and drug-associated PTMs—amino acid distributions and positions [199]. Compared to MS and deep learning-based AI tools (PTM prediction-structure version (MIND-S) framework), the crosstalk between different PTMs, including phosphorylation on HSP90 and their role in drug binding, was precisely predicted by AI [200]. Thus, training data for PTM crosstalk on stress- and yield-associated proteins in polyploids can advance their improvement; however, to avoid false negatives in training data, a combination of AI model prediction and experimental data is recommended for precise outcomes.

Under different external stimuli, a single protein can develop the same or different kinds of PTMs, interacting with different target motifs [36]. For example, different temperature stresses activated phosphatase and kinase and produced different phosphofroms of a target protein in wheat [122]. However, progress in allopolyploid proteomics for stress-related proteoform identification lags far behind compared with other plants and human proteomics due to the inefficiency of the current proteomic approaches. Unlike for other diploids or humans, the advanced computational algorithm of the proteomics (e.g., label free proteomics) often fails precise quantification of stress associated proteins from the multiple paralogus genes of polyploids that encode > 90% identical amino acids and to identify protein variation in a specific locus due to allelic diversity [201,202]. It is also associated with incorrect or incomplete gene models, lack of annotations [203], and lack of reliable bioinformatic tools [202].

Protein enrichment protocol is also important to identify proteoforms, as 1 to 5% proteoforms of the total protein in wheat can be identified using a protein enrichment protocol [204]. However, with necessary modifications, adapting nano-particle enrichment MS proteomics used in human proteomics [205] can improve the proteoform identification in allopolyploids. Furthermore, despite having poor reproducibility and limited ability to detect low-abundance and low-molecular-weight proteins precisely, 2-DE coupled with other proteomic approaches, such as LC-MS/MS, can identify proteoforms (Table 2) [206]. In addition, 2-DE can identify the isoelectric point and relative mass of proteoforms combining with other approaches, such as Orbitrap MS, iTRAQ, TMT, and stable isotope labeling of amino acids in cell-culture (SILAC) [206,207], sometimes used in plant proteomics. An advanced proteomic technique, DIA, can quantify proteins and PTMs precisely, but it requires sophisticated software to overcome the data complexity challenges [208]. DIA-MS identifies peptides using spectral libraries originating from DDA-MS or DIA-MS dataset search or deep-learning models and protein sequence databases (Table 2) [209]. Therefore, the identification of unknown (not listed in the genomic or protein databases) multiple proteoforms with closer structures and unique peptides is challenging in DIA [210]. Recently, DIA has been advanced due to its incorporation into Orbitrap (a high-resolution mass analyzer) [208]—mostly used for DDA—trapped ion mobility spectrometry [211] and high-field asymmetric waveform ion mobility spectrometry [212], but rarely applied in plant proteomics. Another revolutionary DIA is DIA-PASEF, which has significantly seen an increase in its proteomic depth (83% than the previous version), experimental speed and precision, and reduced experimental sample amounts [213]. Improved algorithms of DIA-NN and Spectronaut software (Spectronaut 20: https://biognosys.com/software/spectronaut, accessed on 3 October 2025) in library-free methods lead to more accurate protein and peptide quantification and data consistency than other methods, such as library-based methods [22]. DIA-NN and Spectronaut identified 120.2% and 69.8%, and 30.2% and 11.6% higher peptides and proteins than the library-based methods, respectively. Furthermore, with optimization, this method can be adapted for improved protein quantification from the complex polyploid tissues.

In addition, sample preparation methods and different proteomic approaches influence true proteoform identification. Several recent review articles have discussed the technical challenges of using top-down and bottom-up approaches for proteoform identifications [214,215,216,217,218]. Though top-down proteomics with MS identifies intact proteoforms [214], it faces the challenge of a proteoform mass limit of 30 kDa [219], and requirements of compulsory reproducible samples and a high level of sample fractions [220], resulting in difficulties in finding exact localization [221]. However, improved liquid/gas-phase separation of MS instruments and bioinformatic tools [221]—e.g., TDPortal, Proteoform Suite and MetaMorpheus—alongside false-discovery rate determination, proteoform annotations and databases, and proteome validation [220] are required to overcome the challenges. “Super charging”—forcefully bringing a single charge state of all the molecules of a proteoform within the mass range of the MS instrument—and modified 2-DE-PAGE improve the sensitivity of the top-down approach for higher proteoform coverage [215]. In contrast, though bottom-up proteomics is a more sensitive, faster, and easier approach than top-down proteomics to identify high- and low-abundance proteoforms [206,217], it has limitations in identifying complete proteform information and distinguishing between multiple proteoforms originating from a single protein due to protein digestion and a lack of proteoform database [220,222]. SWATH-MS, and a combination of bottom-up and 2-DE approaches, can be useful for high-resolution proteome identification in bottom-up proteomics [217].

Data reproducibility is still a key challenge for proteomics. Though addressing the challenges in human proteomics has been initiated, it remains under-represented in polyploid proteomics. In human proteomics, reproducibility and data accuracy in DIA-MS have been improved by short-time data acquisition from different instruments in triplicate [223]. Furthermore, a pipeline (ProNorM), developed using computational modules, mitigates instrumental variations, resulting in high reproducibility and statistical analysis potential. On the other hand, DDA’s reproducibility can be improved by a dual-search approach—a spectral library is built using the searched sequence data, and further the same data is searched in the library [224]. This method also minimizes the requirement of large database size, but two different sizes libraries result in significant differences in protein and peptide identification. Improved software tools are required for the practical use of the proposed approach (Table 2). Though the progress of top-down and bottom-up proteomics for proteform identification techniques is mostly explored in animal proteomics, with an improved protein extraction protocol, complete protein and proteoform databases, and bioinformatic tools, they can be explored in polyploid proteomics.

The protein extraction protocol from allopolyploid tissues is still being optimized. The complex cell wall composition and presence of secondary metabolites can hinder complete protein extraction [40,225]. For example, wheat tissues have polysaccharides, lipids, polyphenols, and other secondary metabolites [226], while cotton fibers contain polyphenols and lipids in the cell wall [53]. An improved TCA-B (trichloroacetic acid/acetone-borax/PVPP/phenol) protocol of protein extraction from polysachharide and polyphenol rich tissues for further 2-DE and LC-MS/MS has been proposed [227]. Combination of two different protein extraction methods with an additional phenol extraction yielded high-quality protein from cotton leaves. A combination of multiple protein extraction methods also facilitates the achievement of greater protein coverage. Using a combination of two protein extraction buffers, isoelectric focusing and phenol, and pairing pressure cycling technology for further label-free quantification yielded 66 more proteins than the commonly used phenol extraction method in cotton fiber [228]. Protein extraction buffer and digestion enzymes can be modified according to the crops and tissues used for proteomics. Testing seven different protein extraction protocols in polyploid grains, including wheat and barley, Tris-HCL or urea-based extraction buffer coupled with trypsin-based membrane digestion with filter-aided sample preparation (FASP) found effective for a higher number of protein and peptide identifications than the conventional method [229]. However, trypsin digestion is limited to a shortage of lysine and arginine cleavage sites, often restricting precise proteoform identification. On the contrary, multiple enzymatic digestions—trypsin and glutamyl endoproteinase—improve grain and metabolic proteins identification in wheat [230]. High-quality single-seed storage protein extraction from oilseed Brassica—*B. rapa*, *B. nigra*, *B. juncea*, and *B. fruticulosa*—for 1D-SDS-PAGE based proteomics was improved by using a ball grinder with stainless steel beads in 2.0 mL Safe-Lock tubes with rounded bottoms and hinged lock lids with methanol for 1.2–5.5 mg of single seed grinding [231]. Tris-glycine- and tris-tricine-based extraction buffers were found to be efficient for high- (25–120 kDa) and low- (2–25 kDa) molecular-weight protein separation. Thus, optimized protein extraction protocols are crucial for successful proteomic studies in allopolyploids [40,53,231], while the combination of different protein extraction strategies could be more efficient than a single method.

Another associated challenge is less precise and incomplete protein annotations. Like other diploids, annotation databases of allopolyploids are often incomplete [41,45]. Despite significant advancement of next-generation sequencing, the repetitive genomes, gene duplications, transposable elements and homeologous genes of allopolyploid often challenges genomic annotation of the crops [232]. Although several genomic databases are available for allopolyploid crops, such as Wheat@URGI [233], Bolbase [234], BRAD [235], and CottonGen [236], they often fall short of providing a comprehensive match between transcriptomic and proteomic data. The Plant Public RNA-seq database (PPRD, http://ipf.sustech.edu.cn/pub/plantrna/, accessed on 3 October 2025), a comprehensive web-based platform, offers a common RNAseq libraries for crops, including wheat and cotton, reducing the inequality of data quantification resulting from different bioinformatic tools [237]. To understand the molecular mechanisms of stress tolerance and evolution of polyploids, a common reference standard is recommended. In human omics, standard transcriptomic and proteomic reference datasets have been proposed addressing the variabilities of the two omics [238]. However, developing such reference datasets is challenging for polyploids due to their large genome size with homeologous and duplicated genes, i.e., genetic redundancy, transposable elements, higher level of heterozygosity and chromosomal translocations, and lack of bioinformatic tools for differentiating nearly identical homologs and homeologs and species-specific standard genome assembly [239,240].

Inadequate annotation restricts the identification of protein localization, PTMs, and protein variants in allopolyploids like wheat [225]. Multiomics databases—Wheatomics, BnIR and COTTONOMICS for wheat [241], *B. napus* [242] and cotton [243], respectively—offer better annotation, while plant stress proteome database (PlantPReS; www.proteome.ir) has >10,600 unique stress proteins. However, individual high-quality protein databases remain essential. For example, the wheat proteome database remains incomplete [40,244], limiting our ability to investigate complex, polygenic traits like abiotic stress tolerance [54], organelle-specific proteomics [chloroplast [245], mitochondria [246] and transmembrane [247], and single-cell proteomics [248]. Incorporating the information of uncharacterized proteins and protein–protein interaction, improving conceptual biasness and computational tools, and integration of different databases can improve the protein annotation dataset of the allopolyploids [249].

Furthermore, while the crop databases are incomplete, deciphering information about unknown proteins, which might play a key role in crop improvement, from the existing protein databases is challenging. However, identifying protein function through experiments is expensive and time-consuming, while computational methods can predict it, which is often less effective, as small changes in amino acid sequences can affect protein structure and functions [250]. Deep learning models such as AlfaFold [251] can help overcome the limitations of computational methods. Though deep-learning models have been introduced in plant proteomics (e.g., sequence-based protein language), but they are restricted to identifying DNA-binding proteins [252], and they are also expensive. A recently proposed conventional machine learning-based model, the LightGBM-based method (FUJISAN), in human proteomics offers opportunities for structural similarities of protein domains and pocket-forming residues identification to predict the functionality of unknown proteins [250].

As functional diversification and neofunctionalization of gene products (protein) are common in allopolyploids due to polyploidization, developing machine learning based models will be beneficial to characterize unknown proteins in allopolyploids. It will also open the avenue of protein engineering to improve protein functionality for improving stress resilience and yield in allopolyploids. However, protein engineering—modification of the amino acid sequence of a protein by mutation or inserting or substituting new sequences through site-directed mutagenesis or direct evolution methods to alter the structure and function of proteins aiming modified phenotypes [253]—in allopolyploid is challenging. While locus-specific mutagenesis is rare in plant protein engineering [254], regulation of protein isoforms, complex protein–protein interaction, and non-equilibrium homeologous proteins in the allopolyploids [45] may hinder successful protein engineering.

Difficulties in pathogenic protein identification, and lack of optimized in vitro mimicking media as well as in vivo models [255] often limit proteomics to understand the molecular mechanism of biotic stress tolerance of crops. Pathogenic protein extraction complexity from the host cell [256], MS-based instrument’s limitations, including ionization suppression [257] and lack of complete genome sequence of the pathogens [258] are the major challenges of pathogenic protein identification. Consequently, protein abundance comparison becomes difficult due to the disproportionate host and pathogenic proteins [258], leading to challenges in proteoforms and PTMs identification. The challenges could be intensified if new pathogenic strains evolved. Use of advanced proteomic approaches, such as DIA, reducing heterogeneity among the infected cells through improving pathogen inoculation methods, using improved mimicking media, and improving pathogenic annotation databases, can improve the understanding of host–pathogen interactions. For example, a recently optimized in vitro-mimicking media improved the protein dynamics in *Botrytis cinerea* [259], a major pathogen of many crops, including cotton [260]. Protein–protein interaction (PPI) also plays a crucial role in host–pathogen interactions through controlling cellular mechanisms. Recently developed TurboID-based proximity labeling method overcomes the limitations of low efficiency of PPI identification, using the conventional method [261], which has been proposed for a polyploid, potato [262], indicating the prospect of using this method for other polyploids with developing protocols. However, the requirement of wild variety as control, and optimization of biotin addition time and enzyme (Turbo) [261] needs to be considered during protocol development.

Proteomics often fails to contribute to breeding due to a lack of validation of the protein or proteoform biomarkers. Popular validation approach using antibody-based methods such as Western blotting remains extremely challenging [201] due to a lack of strong affinity to the target protein or amino acid residues and the ability to mark only the target, selectivity [263]. While researchers are searching for an efficient protein validation method, using CRISPR/Cas knockdown/knockout of the target-protein encoding functional genes is a promising approach [264]. For proteoform validation, the use of immunoassays [265] and Proteoform-predictor [266] can be explored in polyploids.

## 6. Prospects of Proteomics in Polyploid Breeding

Rapid advances in genome sequencing technologies have increased the availability of gene sequences for most of the crops. However, the identification of protein coding regions in the genome and their functions in crop performance is essential. With large sequence data in databases, geneticists face the challenge of deciphering the protein products of gene expression. Proteomics, the large-scale study of proteins, holds great promise for revolutionizing plant breeding by enhancing our understanding of the genetic factors underlying complex traits in polyploid crops [45]. Despite major progress over the last two decades in understanding genetic and genomic consequences of polyploidy, research into proteomic-level changes in polyploids remains in its early stages. Protein isoforms and PTMs are key regulators of protein function, activity, localization, and interactions. Current research efforts focus increasingly on linking genotype to proteome to phenotype, and proteomics is expected to emerge as a critical field in polyploid breeding over the coming decade.

During flowering and grain filling, ROS causes oxidative stress and reduces yield [267]. These stresses affect photosynthesis and increase photorespiration by altering cellular homeostasis. However, the complex interactions among gene copy numbers, stress-induced expression changes, and the resulting impacts on the transcriptome, translatome, proteome, metabolome, phenotype, and plant fitness are not fully understood. Investigating the effect of gene content and expression changes in the proteome of autopolyploids and allopolyploids is crucial. Identification of biologically important proteins is also important. To achieve a better understanding of the biologically significant low-abundance proteins, researchers often reduce sample complexity or remove high-abundance proteins.

Polyploidy involves whole-genome duplications, which enables newly formed angiosperm polyploids to tolerate extensive genomic restructuring [188]. These large-scale genome alternation within just one or a few generations result in significant changes to the transcriptome, metabolome, and proteome, ultimately leading to altered phenotypes.

Beyond the evolutionary research challenge, subcellular proteins or multiplex proteins, which are often located in two or more locations, are important for stress tolerance, identification remains challenging [268]. In wheat, isolating specific organelles from whole tissue complicates subcellular proteomic analyses. As a result, many subcellular proteins—particularly stress-related and housekeeping proteins—remain unclassified. A recent AI-based predictor, pLoc-mPlant, has been advanced to pLoc bal-mPlant to identify subcellular proteins in plants using Gene Ontology (GO) annotations and balanced training datasets [269].

The subcellular location resources in The Crop Proteins of Annotated Location database (https://croppal.org/) can be used to improve the model and develop crop-specific datasets for wheat and canola, but need to include other polyploids, including cotton. To understand the host–pathogen interactions, coupled with crop predictors, pathogen-specific subcellular protein predictors such as Gram-LocEN [270] and FunSecKB2 [271] can be considered. Furthermore, the recently proposed dual-perspective protein profiling in clinical proteomics [272], coupled with bioinformatic tools, can be applied in crops including polyploids to identify novel resistance mechanisms, host–pathogen relationships, and disease protection strategies [273]. Therefore, AI-based predictors can help identify organelle-specific proteins, improving our understanding of stress responses and cellular functions in polyploids.

Proteomics helps understand polyploidization, a complementary tool in crop improvement, particularly in identifying differential proteins and their roles in functional expression and evolution. Studying proteome alterations caused by polyploidization gives breeders insights into developing crops with improved traits, boosting productivity and resilience. For example, proteomic studies on hexaploid *Brassica* identified 452 DAPs with a significant expression bias toward the tetraploid progenitor [274]. In wheat, comparative proteomic analyses across diploid, tetraploid, and hexaploid genotypes revealed that protein expression in hexaploid wheat is influenced not just by individual donor genomes but also by complex interactions among these genomes [275]. Similarly, in cotton, allopolyploidization has led to altered gene expression patterns and diversified proteomic profiles. Hu et al. (2015) [276] reported that 4.4–12.8% of differential protein expression between allopolyploids and their diploid progenitors with A and D genome while over 80% of them were additive in nature. Additionally, proteogenomics—the combination of proteomics, genomics, and transcriptomics—is a highly promising method to refine the gene annotations of these complex polyploids, discover protein coding regions in the genes, and identify novel peptides [277]. For example, using proteogenomics, tissue-specific proteomes including novel proteins and their subcellular locations were identified in the 24 organs of *T. aestivum* [244]. Thus, proteomics, combining with other omics, multi-omics, can provide better insights into the molecular mechanisms of the stress response of the polyploids for improving their stress-resilience.

Allopolyplodization also offers an opportunity for allele-specific proteomics and protein modification at their active sites. Multiple genome combination creates gene variant patchworks, providing gene expression flexibility in genotype–environment interactions, leading to improved stress tolerance compared to their progenitors [278]. Stress-resilient phenotypes often differ by a tissue-specific single amino acid polymorphism, which can be identified by genotype-specific peptides. Recently, Carpentier (2020) proposed protocols for allele-specific protein identification for allopolyploids, which may overcome the reference genome biasness. However, besides the proposed proteomics, a crop- and genotype-specific genomic library will add more precise protein information. On the other hand, subgenomes of the allopolyploids do not generally recombine and the duplicate genes often help to play multiple roles without affecting each other [279]. For example, the gene family of a metal transporter protein class, a natural resistance-associated macrophage protein (NRAMP), regulates cadmium toxicity in wheat without affecting other metal ion transportation [279], while it affects manganese, iron, and cadmium transportation in rice [280]. Therefore, using CRISPR/Cas9 coupled with computational models, engineering of the active sites of the proteins through in vivo targeted mutagenesis could develop stable and target specific phenotypes in allopolyploids [254]. Though genome engineering in allopolyploids has been advanced significantly [281,282], protein engineering has not been started yet. Recently, using an in vivo mutagenesis, a potential lysine binding sites of dihydrodipicolinate synthase enhanced salinity and drought stress tolerance and yield in rice [283].

Comparative proteomic studies have significantly contributed to identifying and validating candidate genes for crop improvement. These analyses provide molecular insights into grain yield increase and stress tolerance mechanisms, helping breeders to develop high-yielding varieties with improved stress resilience. For example, in wheat, Daba et al. (2020) [284] identified 3182 DAPs involved in tillering, spike initiation, and kernel development after anthesis using quantitative proteomics. These proteins are promising targets for boosting yield potential. Moreover, DAPs have been identified in relation to various stress tolerances, including heat [114], salinity [285], cold [286], drought [287], and disease resistance [159,288]. In cotton, proteomic research revealed key proteins associated with high-quality fiber development [53,289], and DAPs linked to drought [290,291], cold [292], and salinity tolerance [293].

Overall, proteomics helps elucidate how environmental stresses modify protein structure in polyploid crops. Future advancements in functional and subcellular proteomics will enhance our understanding of stress response pathways and facilitate the development of resilient crop varieties. Notably, breakthroughs in wheat genomics have greatly supported breeding efforts, and the integration of new stress-tolerant genes into breeding pipelines will accelerate the development of homozygous lines with high genetic gains. As proteomic separation techniques improve, proteomics will become increasingly central to modern crop breeding programs.

## 7. Conclusions

Allopolyploid crops such as wheat, oilseed Brassica, and cotton are vital for global food security and economic growth. However, HS and pathogenic infections drastically reduce grain yield, highlighting the need to improve climate resilience and yield stability. The advancement of proteomic approaches has been utilized to unravel the molecular mechanisms underpinning stress responses in these crops. Among the three allopolyploids, proteomic research in wheat has progressed the most. Across these allopolyploid crops, several proteins—particularly those involved in photosynthesis, ROS scavenging, carbohydrate metabolism, peroxidase activity, and HSPs—and protein isoforms have been commonly associated with HS tolerance without grain yield penalty. On the other hand, the proteins involved in responses to biotic stress vary according to the disease specific pathogens. Nonetheless, proteins linked to photosynthesis, defense mechanisms, calcium signaling, lipid metabolism, and lignin biosynthesis commonly contribute to pathogen resistance in the allopolyploids.

Understanding stress-induced proteins at different growth stages is critical for breeding climate-resistant cultivars. In wheat, for example, upregulated proteins such as RuBisCo activase A, and PEP carboxylase, along with antioxidant enzymes like SOD, CAT, and GSH, contribute to HS tolerance in seedlings. Additionally, proteins related to starch biosynthesis, low-molecular-weight glutenin subunits, and gliadin help maintain grain yield and quality under HS. Despite the promising potential of proteomics to enhance our understanding of HS responses and host–pathogen interactions in allopolyploid crops, several limitations remain. A lack of improved protein extraction methodologies, comprehensive protein annotation databases, bioinformatic tools, and pipelines for investigation into protein isoforms, PTMs, and PPI, along with challenges regarding reproducibility, functional validation of candidate protein biomarkers, and subcellular proteomics, are major challenges of proteomics. Combinations of multiple protein extraction methods can offer better protein coverage than the single extraction method. The inefficiency of the current proteomic approaches in overcoming the genetic redundancy of allopolyploids aiming to identify stress-induced proteins, proteoforms, and peptides is another major challenge. Advanced DIA or double-search DDA used in other biological systems can be adapted for improved proteoform identification and reproducibility. An improved TurboID-based proximity labeling method can decipher the PPI to understand the cellular functions of the crops during stress conditions. CRISPR/Cas, immunoassays, and Proteoform-predictor can be potential proteoform validation methods.

Despite challenges, proteomics has great prospects in polyploid breeding. Proteomics can not only be successfully used to discover the evolution of crops and stress-associated genes but also identify the subcellular localization of the candidate proteins. The integration of AI into the existing MS-based proteomic approaches and other predictive tools are promising for identifying stress-induced proteoforms and their subcellular locations, improving protein databases, understanding the crosstalk between PTMs on stress-associated proteins, and protein engineering. Proteomics coupled with other -omics can also be useful for clear understanding of the stress-resilience mechanisms of the allopolyploids. Moreover, addressing the current challenges and gaps of proteomics will be the key to translating molecular insights into practical tools for stress-resilient breeding programs in allopolyploid crops.

## Figures and Tables

**Table 1 proteomes-13-00060-t001:** Summary of proteomic studies on heat stress tolerance in wheat, cotton, and oilseed *Brassica* during the last five years (2019–2025).

Crop	Species	Genotypes/ Varieties/ Lines	Growth Stages	Treatment Condition	Tissue/Organ	Techniques	Validation Status	Total No. of DAPs/Protein Groups	Unique DAPs/peptides	Common Proteins	Summary	Key Genes/Proteins/Enzymes	Reference
Wheat	*Triticum aestivum* L.	Zhoumai 36	Tetrad stage, binuclear stage, and trinuclear stage	Field condition	Anther	TMT; Peptide identification by nano UPLC–MS/MS	qRT-PCR	2532 DAPs	Twenty-seven, 157, and 2348 DAPs in tetrad, binuclear, and trinuclear stages, respectively		Under HS, in the tetrad, binuclear and trinuclear stages 27, 157, and 2348 DAPs were identified, respectively. HS disrupts different signaling pathways, including MAPK signaling pathway, phosphatidylinositol biosynthesis pathway, starch and sucrose synthesis pathway and flavonoid biosynthesis pathway, due to downregulation of related proteins resulting male sterility.	*TraesCS4B02G193500.2* (thermosensitive male sterile 1), *TraesCS7A02G272400.3*, *TraesCS4B02G193500.2* (thermosensitive male sterile 1)	[28]
		KX3302 (high gluten) and BN 208 (medium gluten)	Maturity	Steam at 100–120 °C	Wheat flour	4D label-free		3783 proteins (2124 DAPs)	2861		Storage protein in wheat grain is associated with celiac disease, an autoimmune disease. Wheat genotypes with high- and medium-gluten have α-gliadin and HMW-GSs, and γ-gliadin and LMW-GSs as main DAPs related to the disease, respectively. Heat stress during dough stage reduces the abundance of these proteins in both genotypes resulting in improved diseases.	N/A	[101]
		Xinchun 9	Maturity	38 °C for 1, 2, 8, and 24 h.	Spike	Data Independent Acquisition (DIA)	qRT-PCR	78,885 peptides	19,503 peptides		Proteins associated with proline synthesis, ABA signaling, HSPs, and MADS TFs play an important role in thermotolerance in wheat.	*TraesCS7D02G483200*	[48]
		Tainong 18	Maturity	Field condition	Leaf	iTRAQ		4411 (139 DAPs)			Grain weight and grain yield increased due to delayed sowing resulting from improved electron transport in photosynthetic channel by upregulation of proteins-related to photosynthesis, antioxidant activities and stress. Delayed sowing increases photosynthetic electron transport associated with proteins such as PsbH, PsbR, and PetB under HS, which supports high photosynthesis to increase HS tolerance.		[102]
		BWL4444 (HD2967+ Yr10; tolerant)	Anthesis	35/24 °C	Grain	2D-PAGE; MALDI-TOF/MS	qRT-PCR	153 DAPs			Day–night HS suppresses stress tolerant gene expression due to activation of multiple pathways for proteins associated with protein translation, gliadins, low-and high- molecular weight glutenin, glycolysis, photosynthesis and defense. Downregulation of proteins associated with photosynthesis, glycolysis, metabolic pathways and improper protein folding increase HS sensitivity and decreases grain weight, yield, and quality.		[103]
		HD3086 (tolerant) and BT-Schomburgk (sensitive)	Flowering and grain development stages	32 °C and 38 °C for 2 h	Leaf	2D-PAGE; MALDI-TOF/MS	N/A	Approximately 23,000 DAPs	Twenty-two and 11 unique DAPs in tolerant and sensitive varieties, respectively	Twenty	Mitogen-activated protein kinase (MAPK) positively increases HS tolerance by increasing proline, H_2_O_2_, gene expression of antioxidant enzymes, defense and osmolytes and HSPs. MAPK also regulates grain quality under HS.	MAPK genes	[104]
		Chinese Spring	Grain filling	30 °C and 40 °C	Grains	Tandem mass tags (TMT)	N/A	3742 (297 and 461 DAPs at 30 and 40 °C, respectively)		150	Posttranscriptional regulation plays an important role in mild HS (30 °C), while amino acid frequency is important for protein expression during severe HS (40 °C). Under HS, a higher frequency of the AAG codon was found to alter HS-induced protein expression. Furthermore, though ribosomal protein influenced HS induced proteins, no transcriptional regulation was evident under 30 °C.		[105]
		Unnat Halna	Seedling	38 °C for 4 h	Leaf	2-DE	Vacuum-assisted *Agrobacterium* transformation method	55 DAPs (10 and 29, upregulated and downregulated, respectively)			Heat stress-induced 2-cysteine peroxiredoxin (2CP) increases HS tolerance by increasing chlorophyll a and b content and decreasing H_2_O_2_ and ROX concentrations. The 2CP in wheat also interacts with protochlorophyllide reductase b, TaPORB, and plays an important role in chlorophyl biosynthesis.	*Ta2CP*	[106]
		HD2985 (tolerant) and HD2329 (sensitive)	Pollination and grain filling	38 °C for 2 h	Leaf, stem, and spike	iTRAQ	qRT-PCR	318 groups	17 DAPs associated with stress associated active proteins	9425 (3600 and 5825 upregulated and downregulated, respectively)	Heat shock proteins including HSP20, HSP26, and HSC70; signaling molecules including MAPKs and CDPKs; and antioxidant enzymes, including SOD, CAT, and APX play an important role in HS tolerance. On the other hand, upregulated oleosin, globulin 3A, 3B, and gamma gliadin, and α/β amylases regulate grain quality of wheat under HS.	Carboxylase enzyme	[107]
		Gaocheng 8901	Flowering to maturity	40 °C for 2 h	Milled grain	iTRAQ; LC-ESI/MS	qRT-PCR; immunobloting	207 DAPs			Kernel weight and starch content in wheat decreases due to protein–protein interaction of HS-induced proteins, including elicitor responsive gene 3, brassinosteroid-insensitive 1, chaperone protein, histone cell cycle regulator and pre-mRNA processing factor.		[108]
Cotton	*Gossypium hirsutum*	GH-Hamaliya (tolerant) and GH-Hamaliya (susceptible)	Flowering	45/32 °C	Leaf	Label-free	N/A	8005	494	13	During HS, photosynthesis decreases due to a decrease in intercellular CO_2_ concentration, stomatal conduction and water status; chlorophyll reduces due to decreased lipid peroxidation of thylakoid and chloroplast membrane. However, increased proline content supports HS-tolerance. More than 8,000 DAPs were identified under HS. Upregulation of proteins associated with ROS scavenging, ATP synthesis, and major latex-like proteins, beta-glucosidase and HSP play an important role in HS-tolerance.		[109]
		TM-1 and W0	Seedling	42 °C for 24 h	Leaf	SDS-PAGE	Yeast two-hybrid assay for protein–protein interaction	2426	728	Three	Reduction in expression of GhHSP24.7 plays an important role in heat and drought stress tolerance through increasing ROS scavenging, decreasing acetylation, stomatal conductance and H_2_O_2_ content.	*GhHSP24.7*	[110]
		Sicot 71	Reproductive	40/30 °C for five days	Leaf	SWATH-MS		880	728	Two	In mature pollen, HS suppresses translational processes by disturbing metabolic processes, such as starch biosynthesis, glycogen catabolism, and xyloglucon processes. However, transcription remains active. In all pollen development stages, HSPs and peptidylprolyl isomerase were common.		[50]
	*Gossypium hirsutum*	Sicot 71	Tetrads and binucleate stage	36/25 °C and 40/30 °C for five days	Pollen grain	Label-free		876	106 DAPs		Heat stress reduces the reproductive capacity of pollen grain of cotton by decreasing the pollen grain size at the tetrad stage, increasing sugar percentage in the pollen grain, and decreasing pollen viability. Upregulation of HSPs, including HSP70 kDa proteins related to starch and sucrose metabolism, and downregulation of late embryogenesis abundant proteins contribute to the HS tolerance in cotton’s pollen.		[111]
	*Gossypium hirsutum*	Sikang 3	Bolling stage	38 °C and 40% field capacity	Boll shell	2D-PAGE; label-free quantitation	3094 and 3049		2927		Under drought and HS, *Bt* gene expression in cotton increased starch hydrolysis ability and decreased fructose and glucose content. Proteins associated with protein export pathways, particularly signal recognition particles, play a crucial role in restricting the transportation of nascent peptide chain into the endoplasmic reticulum due to HS, resulting in decreased Bt-insecticidal protein in cotton.		[112]
	*Gossypium robinsonii*	Australian wild cotton	Pollen development stages (tetrad, uninucleate and binucleate)	36/25 °C and 40/30 °C for five days	Pollen grain	Nano LC-MS/MS; IDA-DIA SWATH-MS	2704	422, 489 and 94 DAPs at 36 °C and 297, 157 and 61 DAPs at 40 °C in tetrad, uninucleate and binucleate, respectively	196 DAPs	36 and 11 at 36 and 40 °C, respectively	Heat stress at the late stage of pollen development reduces translational response and protein abundance. Plasmodesma related proteins play an important role in protein transport and cell–cell communication at the tetrad stage of pollen under HS. Downregulation of Rab proteins plays a crucial role in thermotolerance by inhibiting protein transport.		[113]
Brassica	*Brassica juncea*		Seedling	45 °C for 4 and 8 h	Leaf	MALDI-TOF-MS		22954 DAPs	81 DAPs	Differentially accumulated proteins: 77 at 4 h, 81 at 8 h, 74 at both 4 and 8 h	Under 4 h and 8 h of HS, 119 and 81 DAPs, 532 and 570 peptide sequences were identified, respectively. During 8 h HS, accumulation of HSP, including HSPs and HS-induced TF A-4a increased. On the other hand, during 4 h, MuTL like-protein-1, Cytochrome P450 85A1, CDT1_C domain-containing protein and B3 domain containing proteins, putative calcium-transporting ATPase 11 and an uncharacterized protein accumulation increased, and ribosome inactive protein accumulation decreased.		[114]
	*Brassica juncea*		BPR 543-2	37 °C, 3 h interval for five days	Pericarp	Label-free	1078	209 DEP	78	N/A	Under HS and a combination of salt-stress and HS, 367 and 209 DAPs, including HSPs, proteins related to cell division, protein disulfide isomerase, calcium binding proteins and protease, were highly accumulated, respectively.		[115]
	*Brassica campestris*	WS-1	Reproductive stage	40/30 °C	Leaf	TMT	qRT-PCR	2806 DAPs	1787 DAPs	1022	Glucose transporter gene, *BccrGLU1*, improves HS-tolerance by increasing ROS scavenging, and reducing glutathione content and ratio of glutathione/oxidized glutathione.	*BccrGLU1*	[116]

N/A = not applicable; TMT = tandem mass tag; UPLC = ultra-performance liquid chromatography; MS = mass spectrometry; 4D = four-dimensional; iTRAQ = isobaric tags for relative and absolute quantitation; 2-DE = two-dimensional gel electrophoresis; SDS-PAGE = sodium dodecyl sulfate–polyacrylamide gel electrophoresis; LC = liquid chromatography; ESI = electrospray ionization; SWATH = sequential window acquisition of all theoretical mass spectra; MALDI = matrix-assisted laser desorption/ionization; TOF = time-of-fight; qRT-PCR = quantitative real-time reverse-transcription–polymerase chain reaction; DAPs = differentially abundant proteins; TF = transcription factors; HS = heat stress.

**Table 2 proteomes-13-00060-t002:** Advantages and limitations of the major proteomic methods for allopolyploid proteomics.

Proteomic Techniques/Methods	Advantages	Limitations
2DE-MS (2DE-PAGE; 2D-DIGE)	Simple and easy to use High resolution proteome coverage coupled with advanced techniques	Low reproducibility Less effective to identify low abundant and low molecular weight proteins
Data dependent acquisition (DDA: Label-free)	Precise quantification of all peptides of a sample High reproducibility	It is inefficient to identify extensive low abundance proteins or peptides.
Data independent acquisition (DIA: SWATH-MS)	Ability to identify protein and proteoforms precisely Higher data consistency than other methods High data reproducibility	Identification of unknown proteins is challenging Dependent on DDA-MS for spectral library Sophisticated software is required for precise peptide identification

2DE-PAGE = 2-dimensional polyacrylamide gel electrophoresis-MS; MS: mass spectrometry and 2D-DIGE: 2-dimensional difference gel electrophoresis.

## Data Availability

No new data were created or analyzed in this study.

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
