# Peer review of "Proteomics in Allopolyploid Crops: Stress Resilience, Challenges and Prospects"

_proteomes, 2025, doi:10.3390/proteomes13040060_

Round 1
Reviewer 1 Report
Comments and Suggestions for Authors
In my opinion, the manuscript would benefit from including the following additions. The authors may revise accordingly based on the content below:
In the section—Genome complexity of polyploidy
The genome of allopolyploid crops (such as wheat, cotton, and rapeseed) is a dynamic, non-equilibrium, and highly interactive system rather than a static "patchwork." Its complexity lies not only in the inclusion of multiple sets of genomes but also in the profound and intricate interactions, rearrangements, and co-evolution that occur among these combined genomes. These dynamic processes provide allopolyploid crops with tremendous genetic and expressional plasticity, enabling them to adapt to new environments and be domesticated for traits beneficial to humanity. So, the complexity of polyploid crop genomes severely constrains research in their genomics and genetics, while proteomics, which directly investigates protein molecules closely linked to phenotypes, may present a significant opportunity for helping polyploid crop studies.
In the section—Challenges of proteomics in polyploid crops
Currently, the major bottleneck in proteomics is that researchers often merely utilize it as a tool for identifying superior genes. For instance, a genetic modification (DNA level) enhances the expression of its own protein or downstream proteins, thereby improving traits such as heat tolerance or disease resistance. If we analogize DNA as the commander of an army and each ultimately phenotype-determining protein as a soldier, current scholars focus more on how to strengthen the commander and improve their strategic deployment skills, rather than further enhancing the individual combat capability of each soldier. I believe that for proteomics to advance and become more meaningful and prominent in crop research, it is essential to improve protein (proteins with highly induced expression) functionality, particularly for those proteins that directly determine phenotypes. For example, methods such as re-engineering the active sites of heat shock proteins to make them more potent than before should be explored.
In the section—Prospects of proteomics in polyploid breeding
Due to the greater genomic fault tolerance of polyploid genomes, editing functional protein sites in such systems presents distinct advantages. For example, through gene editing of the Fe, Mn, and Cd ion channel protein (encoded by the NRAMP5 gene), the modified channel selectively absorbs Fe while excluding ions such as Mn, and Cd. Notably, this editing approach enhances crop yield only in polyploid backgrounds, where complete functional copies of the ion channel are retained.
Author Response
We thank the reviewer for the critical review of the manuscript and for useful suggestions to improve the manuscript. In the manuscript, we have incorporated the suggestions in the manuscript accordingly, and highlighted yellow in the text.
(1) In the section—Genome complexity of polyploidy
The genome of allopolyploid crops (such as wheat, cotton, and rapeseed) is a dynamic, non-equilibrium, and highly interactive system rather than a static "patchwork." Its complexity lies not only in the inclusion of multiple sets of genomes but also in the profound and intricate interactions, rearrangements, and co-evolution that occur among these combined genomes. These dynamic processes provide allopolyploid crops with tremendous genetic and expressional plasticity, enabling them to adapt to new environments and be domesticated for traits beneficial to humanity. So, the complexity of polyploid crop genomes severely constrains research in their genomics and genetics, while proteomics, which directly investigates protein molecules closely linked to phenotypes, may present a significant opportunity for helping polyploid crop studies.
-Response:
We have added relevant information in the “Introduction” section (L129–143).
(2) In the section—Challenges of proteomics in polyploid crops
Currently, the major bottleneck in proteomics is that researchers often merely utilize it as a tool for identifying superior genes. For instance, a genetic modification (DNA level) enhances the expression of its own protein or downstream proteins, thereby improving traits such as heat tolerance or disease resistance. If we analogize DNA as the commander of an army and each ultimately phenotype-determining protein as a soldier, current scholars focus more on how to strengthen the commander and improve their strategic deployment skills, rather than further enhancing the individual combat capability of each soldier. I believe that for proteomics to advance and become more meaningful and prominent in crop research, it is essential to improve protein (proteins with highly induced expression) functionality, particularly for those proteins that directly determine phenotypes. For example, methods such as re-engineering the active sites of heat shock proteins to make them more potent than before should be explored.
-Response:
We have added relevant information in the “Challenges of proteomics in polyploid crops” section (L776–786).
(3) In the section—Prospects of proteomics in polyploid breeding
Due to the greater genomic fault tolerance of polyploid genomes, editing functional protein sites in such systems presents distinct advantages. For example, through gene editing of the Fe, Mn, and Cd ion channel protein (encoded by the NRAMP5 gene), the modified channel selectively absorbs Fe while excluding ions such as Mn, and Cd. Notably, this editing approach enhances crop yield only in polyploid backgrounds, where complete functional copies of the ion channel are retained.
-Response:
We have added relevant information in the “Prospects of proteomics in polyploid breeding” section (L883–903).
Reviewer 2 Report
Comments and Suggestions for Authors
The review fails to focus and deliver on the title. the authors want to discuss many different and general proteomics topics and focus on heat stress. This makes that the review is very long lacks a general overview and losses the reader in irrelevant details while important challenges and development on genome prediction, mRNA seq developments for polyploids and the usage of those as a database to search against are not discussed. Also the huge development in software allowing DIA and the challenges still around this for polyploids are not discussed. The review needs to be shortened substantially and focus more on the literature and solutions and methods developed for polyploid crops. There are sufficient reviews on heat stress and crops in general. The number of citations is also too high. Try to limit and focus on the ones that are really relevant for polyploid crops and extend beyond cotton, wheat and brassica.
Detailed remarks:
Figure 1 is a very general figure and totally not specific for polyploid crops. The challenges are unsequenced genomes, translocated chromosomes and the redundancy in homologous genes. The authors should more focus on those solutions.
633:the link with CRISPR is absolutely not clear. Also the usage of IA in protein annotation and 3D structures needs to discussed more in detail. Great progress have been made there and annotation is a major bottleneck in polyploid crops!
639:the authors rightfully state inefficiency in current proteomic approaches but should explain better why they are inefficient and what are the specific challenges for polypoids!
649:DIA adds more challenges to identify the proteoforms if they are not predicted from a database. Discuss
651:several other hardware adaptations from other vendors are worth mentioning. Why only focus on orbitrap? The biggest revolution was the development of new software like DIANN and spectronaut. Discuss.
Author Response
The authors thanks to the reviewer for constructive comments and suggestions. We have addressed the comments and highlighted yellow in the text.
(1) The review fails to focus and deliver on the title.
-Response:
We have revised the title as “Proteomics in allopolyploid crops: Abiotic and biotic stress resilience, challenges and prospects”
(2) The authors want to discuss many different and general proteomics topics and focus on heat stress. This makes that the review is very long lacks a general overview and losses the reader in irrelevant details while important challenges and development on genome prediction, mRNA seq developments for polyploids and the usage of those as a database to search against are not discussed.
-Response:
Besides challenges and prospects of proteomics for allopolyploids, the manuscript focuses on the application of proteomics to study heat stress and disease associated with the three major allopolyploids. Additionally, to support the focus, we have discussed a few general features of proteomics. In the revised manuscript, to improve the reader’s interest, we have revised/removed some less relevant details.
We have highlighted the suggested challenges, genome prediction and the importance of mRNA-seq data for proteomic study of polyploids in the “Challenges of proteomics in polyploid crops” section in L733–736 and L739–750, respectively.
(3) Also the huge development in software allowing DIA and the challenges still around this for polyploids are not discussed. The review needs to be shortened substantially and focus more on the literature and solutions and methods developed for polyploid crops.
-Response:
The challenges of the DIA for polyploid study and the relevant advanced software to improve DIA’s performance have been discussed in the “Challenges of proteomics in polyploid crops” section in L647–650 and L654–661. We believe that this information has improved the manuscript focus on the existing challenges of the proteomic approach for polyploid studies and provide directions to overcome those challenges.
(4) There are sufficient reviews on heat stress and crops in general. The number of citations is also too high. Try to limit and focus on the ones that are really relevant for polyploid crops and extend beyond cotton, wheat and brassica.
-Response:
Though there are a couple of reviews discussing proteomics on heat stress in crops, but there are still no reviews that specially focus on the targeted allopolyploids. Additionally, our manuscript comprehensively discusses the advancement of proteomic approaches to identify heat stress associated with proteins and proteoforms, posttranslational modifications in three major allopolyploids, wheat, cotton and brassica oilseed. Therefore, we believe that this review will add value to the polyploid breeders to use proteomics for improving heat stress-resilience in the target crops.
We have added information beyond wheat, cotton and brassica (L258–261, L276–277, L403–405).
(5) Figure 1 is a very general figure and totally not specific for polyploid crops.
-Response:
As the figure was developed based on the proteomic approaches commonly used in the target crops, we have revised the figure title as “Common proteomic approaches used in allopolyploid crops:”
(6) The challenges are unsequenced genomes, translocated chromosomes and the redundancy in homologous genes. The authors should more focus on those solutions.
-Response:
The challenges were discussed in L575–577. We have added more information in the “Challenges of proteomics in polyploid crops” section in L729–732 and L742–746.
(7) 633:the link with CRISPR is absolutely not clear. Also the usage of IA in protein annotation and 3D structures needs to discussed more in detail. Great progress have been made there and annotation is a major bottleneck in polyploid crops!
-Response:
The sentence has been revised as “Use of AI in identifying critical PTMs has been significantly advanced in human proteomics.” (L613) for better clarity.
The use of AI in protein annotations and protein structure has been added in L613–624, L759–782 and L763–774.
(8) 639:the authors rightfully state inefficiency in current proteomic approaches but should explain better why they are inefficient and what are the specific challenges for polypoids!
-Response:
We have explained the reasons of the inefficiency of the proteomic approaches and specific challenges associated with the polyploid study in the “Challenges of proteomics in polyploid crops” section in L630–635.
649:DIA adds more challenges to identify the proteoforms if they are not predicted from a database. Discuss
-Response:
We have discussed the challenges associated with the DIA in the “Challenges of proteomics in polyploid crops” section in L648–652.
651:several other hardware adaptations from other vendors are worth mentioning. Why only focus on orbitrap? The biggest revolution was the development of new software like DIANN and spectronaut. Discuss.
-Response:
We have discussed advanced software associated with the DIA in the “Challenges of proteomics in polyploid crops” section in L656–662.
Reviewer 3 Report
Comments and Suggestions for Authors
Interesting but still unfinished manuscript. As thematic is too broad and complex it is hard to give review that could deep enough describe problems and potential solutions in the area. So it is more informative than educative in nature as it seems to me. Nonetheless, it certainly has its qualities. However, even those qualities are not presented well. As first Table 1 has to many columns that need to be reduced by merging as suggested in PDF file. Also, there is need for second table that will describe the main properties advantages and drawbacks of main proteomic methods.

No special comments.
Author Response
The authors thanks to the reviewer for constructive comments and suggestions. We have addressed the comments and highlighted green in the text. The language of the text has been improved and highlighted in green. All other comments are also highlighted in green.
Responses:
- The names of the developmental stages are mentioned in L215.
- Different reference style corrected L269
- Day-night temperatures reduced thermotolerance, grain yield and quality by restricting the expression of transcription factors (TFs), genes associated with signaling pathway and abiotic stress, and function of protein folding machinery
-Response: Corrected.
(4) Also, there is a need for a second table that will describe the main properties, advantages and drawbacks of main proteomic methods.
-Response: We added the recommended table to the supplementary material and mentioned it in L699–700.
Round 2
Reviewer 3 Report
Comments and Suggestions for Authors
No further big comments, but I think Table 1 should be in main text not as supplement (since this is only thing in the supplements).
Author Response
The authors thank the reviewer for further suggestions. We have addressed the comment and highlighted it in the main text.
Comment: No further big comments, but I think Table 1 should be in the main text, not as supplement (since this is only thing in the supplements).
Response:
We have added the table in the main text as Table 2 (L669 to L671), and cited it accordingly in the text (L643, L650 and L703).